# Grazing resistance developed in *Escherichia coli* K-12 during coexistence with a bacterivorous protist

**Kanji Nakamura** [ORCID] *[☯], **Keisuke Miyauchi**[☯]

Department of Civil and Environmental Engineering, Tohoku Gakuin University, Sendai, Miyagi, Japan

☯ These authors contributed equally to this work.
* knaka@mail.tohoku-gakuin.ac.jp

## Abstract

A development of grazing resistance in *Escherichia coli* K-12 was examined in the presence of a bacterivorous protist, *Spumella* sp. TGKK2. Two transformants were generated from *E. coli* K12 for grazing experiments. One was *E. coli* K-12-TGF, which possesses tetracycline resistance and green fluorescence. The other was *E. coli* K-12-KRF with kanamycin resistance and red fluorescence. These strains can be selectively colonized on antibiotic-containing agar media and further confirmed by their fluorescent colors. First, we added protist-untouched *E. coli* K-12-KRF to protist-touched residual *E. coli* K-12-TGF that had been attacked by *Spumella* sp. TGKK2 in a batch test. Then the survivability of the respective strains was investigated. Consequently, *E. coli* K-12-KRF was predated preferentially. On the other hand, *E. coli* K-12-TGF in the same tube was less predated, indicating some grazing resistance. Similar phenomena were observed when the conditions of these two strains of bacteria were reversed. Also, a continuous culture device supplied with a glucose-containing medium as a substrate was operated. The device connected two complete mixed reactors in series. *E. coli* K-12-TGF was cultivated in the first reactor, and then grown *E. coli* K-12-TGF was predated by *Spumella* sp. TGKK2 in the second reactor. The effluent in the second reactor containing residual *E. coli* K-12-TGF and *Spumella* sp. TGKK2 was supplemented with batch-cultured *E. coli* K-12-KRF. Consequently, it was confirmed that bach-cultured *E. coli* K-12-KRF never exposed to protist was predated preferentially. These findings reveal that *E. coli* K12 acquires some predation resistance through coexistence with the bacterivorous protist.

## Introduction

In the field of bioremediation, the process of introducing particular bacteria cultured elsewhere into contaminated sites to remediate contaminated soil and groundwater is known as bioaugmentation [1, 2]. However, very few introduced bacteria have demonstrated their ability to degrade, leading to complete decontamination. This unsuccessful bioaugmentation is due to the quick reduction of the bacterial population size at the beginning of the augmentation

**Data Availability Statement:** All relevant data are within the paper and its Supporting Information files.

**Funding:** JSPS KAKENHI Grant Numbers 20K21023.

**Competing interests:** The authors have declared that no competing interests exist.

period in cases where bacteria released to the site cannot grow well on pollutants. In a laboratory study, predation by bacterivorous protists has been reported to be the primary cause [3]. Our survey indicated that flagellates close to the genus *Spumella* were the dominant environmental bacterial predators in Japan [4]. It has also been reported that flagellates, closely related to *Spumella*, are the major bacterivorous protists widely distributed in the environment worldwide [5]. Therefore, it is possible that bacteria introduced to a contaminated site for bioaugmentation are predated mainly by a protist that is close to *Spumella*.

Previous studies [6, 7] have shown that bacteria added to environmental groundwater and river water reduce their numbers quickly, presumably by the predation of indigenous protists. In contrast, indigenous bacteria living in those environmental waters were relatively stable without being predated by protists. Thus, bacteria cultured and then introduced to environmental waters were rapidly predated by indigenous protists, whereas indigenous bacteria are less likely to be predated. So, we hypothesized that indigenous bacteria can acquire some grazing-resistant traits by coexisting with protists in the natural environment.

In order to verify this hypothesis, we have performed some experiments using a bacterium, *Cupriavidus* sp. KN1 and a purified bacterivorous protist, *Spumella* sp. TGKK2 [8]. *Cupriavidus* sp. KN1 is a bacterium from the natural environment and can degrade trichloroethylene by its phenol hydroxylase [9]. We examined the conditions needed to acquire the predation resistance using *Cupriavidus* sp. KN1 in the laboratory [8]. Two transformants with different drug resistance were generated based on *Cupriavidus* sp. KN1. In addition, these two strains were prepared under two conditions with and without a history of protist coexistence and mixed to compare their predatory properties. The results revealed that some predation resistance was acquired by protist coexistence.

In this study, prey bacteria were further expanded to clinically derived *Escherichia coli*, not from the environment, for experiments on the formation of predation resistance. We chose well-known and well-studied *E. coli* K-12 as the second model bacterium. If predation resistance formation is observed even in *E. coli* K-12, this phenomenon may be a common function among various bacteria.

## Materials and methods

### Bacteria and a protist used

Bacteria and a protist used in this study are listed in Table 1. Prey bacteria used for the predation experiment are two transformants of *Escherichia coli* K-12, a typical *Escherichia coli* strain. The first transformant is *E. coli* K-12-TGF, which produces a green fluorescent protein (GFP) and is tetracycline resistant (Tc$^r$). The second transformant is *E. coli* K-12-KRF, which produces a red fluorescent protein (RFP) and is kanamycin resistant (Km$^r$). Thus, the two strains possess different drug resistances and can be selectively measured by Colony Forming Unit (CFU) using agar media containing Tc or Km, even when they coexist. In addition, they can

**Table 1. Bacteria and a protist used.**

| Species | | Description | Reference or source |
|---|---|---|---|
| *Escherichia coli* | SM10 λpir | *thi-1 thr leu tonA lacY supE pro recA::RP4-2* Tc::Mu, Km$^r$, *λpir* | [11] |
| | K-12 | Prototroph | ATCC10798 |
| | K-12-TGF | Tc$^r$, ZsGreen | This study |
| | K-12-KRF | Km$^r$, tdTomato | This study |
| *Spumella* sp. | TGKK2 | | [10], NBRC111014 |

be confirmed by their fluorescent colors. The bacterivorous protist is the flagellate *Spumella* sp. TGKK2 (NBRC111014) [10].

LB medium (Bacto^TM tryptone 10 g, Bacto^TM yeast extract 5 g, and NaCl 5 g were dissolved in 1 L of purified water, pH was adjusted to 7.0) was used for cultivating *E. coli* strains. Fifteen g L$^{-1}$ of purified agar powder was added to LB medium to measure colony forming units. The final Tc concentration was adjusted at 5 mg L$^{-1}$ for selective growth of *E. coli*-TGF. The final Km concentration was set at 50 mg L$^{-1}$ for *E. coli*-KRF. The incubation temperature of all *E. coli* strains was 37˚C. Growth of these strains on LB agar mediums containing Tc or Km is shown in S1 Fig.

## Introduction of fluorescent protein genes

The fluorescent protein genes utilized in this study are the green fluorescent protein gene ZsGreen in pZsGreen vector (Clontech Laboratories Inc.) and the red fluorescent protein gene tdTomato in ptdTomato vector (Clontech Laboratories Inc.). As shown in Fig 1, ZsGreen was paired with Tc$^r$ gene, and tdTomato was paired with Km$^r$ gene. Together, they are expressed at the *P1* and *P2* promoters that are transcribed in the opposite direction from the start of Tc$^r$ gene [12]. The drug resistance genes were expressed by the *P2* promoter. In contrast, the

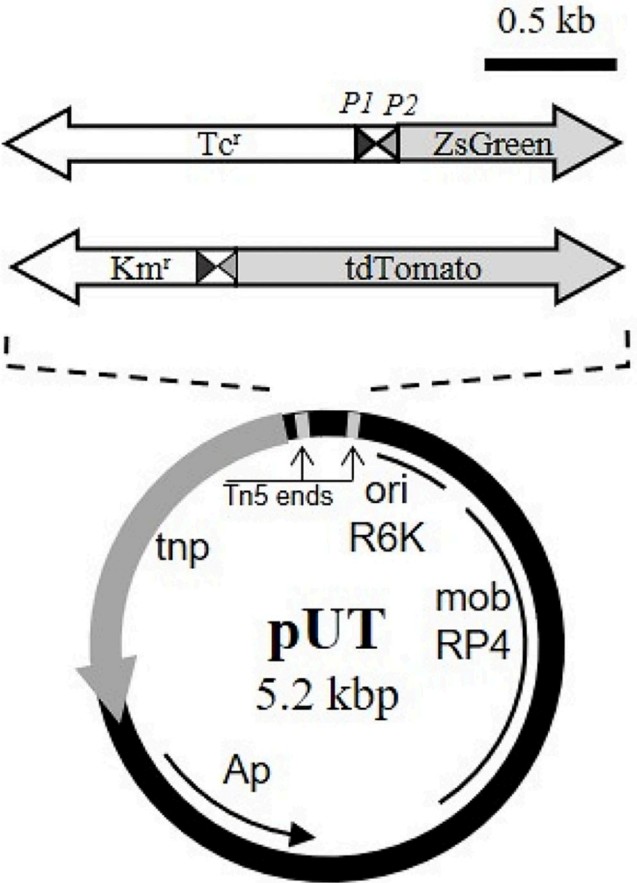

**Fig 1. Integrated genes and transposon vector pUT.** ZsGreen was paired with Tc$^r$ gene, and tdTomato was paired with Km$^r$ gene. They are expressed by the *P1* and *P2* promoters.

fluorescent protein genes were expressed by the *P1* promoter. These genes were inserted into the multiple cloning site (MCS) of Tn5 transposon vector pUT [11]. Then *E. coli* SM10 λpir was transformed by this recombinant pUT. This transformant and *E. coli* K-12 were conjugated overnight on LB agar medium at 37˚C. The conjugated bacteria were diluted as appropriate and applied to an agar medium for the selection of a transformant. This agar medium was prepared by adding 20 mM of sodium lactate and 5 mg $L^{-1}$ of Tc to nutrient broth soyotone yeast extract inorganic basal medium (hereinafter referred to as NSY-IB medium) that contained only inorganic compounds [75 mg of $MgSO_4 \cdot 7H_2O$ $L^{-1}$, 43 mg of $Ca(NO_3)_2 \cdot 4H_2O$ $L^{-1}$, 16 mg of $NaHCO_3$ $L^{-1}$, 5 mg of KCl $L^{-1}$, 3.7 mg of $K_2HPO_4 \cdot 3H_2O$ $L^{-1}$, 4.4 mg of $Na_2EDTA$ $L^{-1}$, 3.2 mg of $FeCl_3 \cdot 6H2O$ $L^{-1}$, 1.0 mg of $H_3BO_3$ $L^{-1}$, 0.2 mg of $MnCl_2 \cdot 4H2O$ $L^{-1}$, 0.02 mg of $ZnSO_4 \cdot 7H_2O$ $L^{-1}$, 0.01 mg of $CuSO_4 \cdot 6H_2O$ $L^{-1}$, 0.01 mg of $CoCl_2 \cdot 6H_2O$ $L^{-1}$, 0.006 mg of $Na_2MoO_4 \cdot 2H_2O$ $L^{-1}$, 0.1 mg of $NiCl_2 \cdot 6H_2O$ $L^{-1}$; pH 7.2] [13]. Thus, the transformant *E. coli* K-12-TGF was selected. Similarly, *E. coli* K-12-KRF was obtained using the same agar medium containing 50 mg $L^{-1}$ of Km.

## Batch cultures of bacteria

The prey bacteria, *E. coli* K-12-TGF and *E. coli* K-12-KRF, were cultured with LB medium at 37˚C and shaken at 200 rpm overnight. Cultures were washed twice with NSY-IB medium (11,100×g, 5 min, 4˚C. The supernatant was discarded and suspended in NSY-IB medium) and resuspended in NSY-IB medium. They were subsequently used after shaking at 20˚C and 200 rpm for a day.

## Continuous cultures of a bacterium and a protist

A continuous culture device consisting of two complete mixed reactors connected in series (Fig 2) was operated for 31 days. The substrate supplied was 200 mg $L^{-1}$ of glucose-containing NSY-IB supplemented with 75 mg $L^{-1}$ of $NH_4Cl$, 100 mg $L^{-1}$ of $Na_2HPO_4$, and 150 mg $L^{-1}$ of $KH_2PO_4$. Reactor workingvolume was 100 mL. The hydraulic retention time (HRT) was set at 30 h (Q = 3.33 mL $hr^{-1}$). The temperature was adjusted at 20 ± 1˚C. The agitation in the vessel was conducted by a stirrer bar rotated at 400 rpm. The first reactor was a bacterial culture chamber for *E. coli* K-12-TGF in which glucose in the substrate was degraded, and the bacterium grew. The second reactor was a protistan culture vessel in which grown *E. coli* K-12-TGF was predated by *Spumella* sp. TGKK2. The effluent from the first tank was collected from sampling port 1, and the effluent from the second tank was collected from sampling port 2. Then glucose, TOC, pH, and CFU were measured. For the effluent from the second reactor, protistan numbers were also directly counted by microscopy using Toma's hemocytometer. Glucose concentrations were measured using the Glucose Assay Kit-WST (Dojindo Laboratories Co.). The measurement method followed the company's manual. The detection limit of the glucose assay kit was 3.6 mg $L^{-1}$, as shown in the manual. TOC analyzer TOC-$V_{CHS}$ (Shimadzu Co.) was used for TOC measurement. Two types of TOC were measured to grasp the various organic matter levels in the tank. One was all TOC, including bacteria and protists, and the other was TOC of the supernatant (supernatant TOC) obtained by precipitating turbid substances such as bacteria and protists by the centrifugation (11,100 x g, 10 min, 4˚C). After the centrifugation, the protist concentration was confirmed to be lower than 1 x $10^4$ cells $mL^{-1}$ using Toma's hemocytometer. The difference between all and supernatant TOC was defined as suspended TOC consisting of bacteria, protists, and their debris.

## Grazing resistance evaluation

In experiments with batch-cultured bacteria, *E. coli* K-12-KRF was fed on *Spumella* sp. TGKK2. Three different initial concentrations ($2.4 \times 10^6$, $2.4 \times 10^7$, $2.4 \times 10^8$ CFU $mL^{-1}$) of *E. coli*

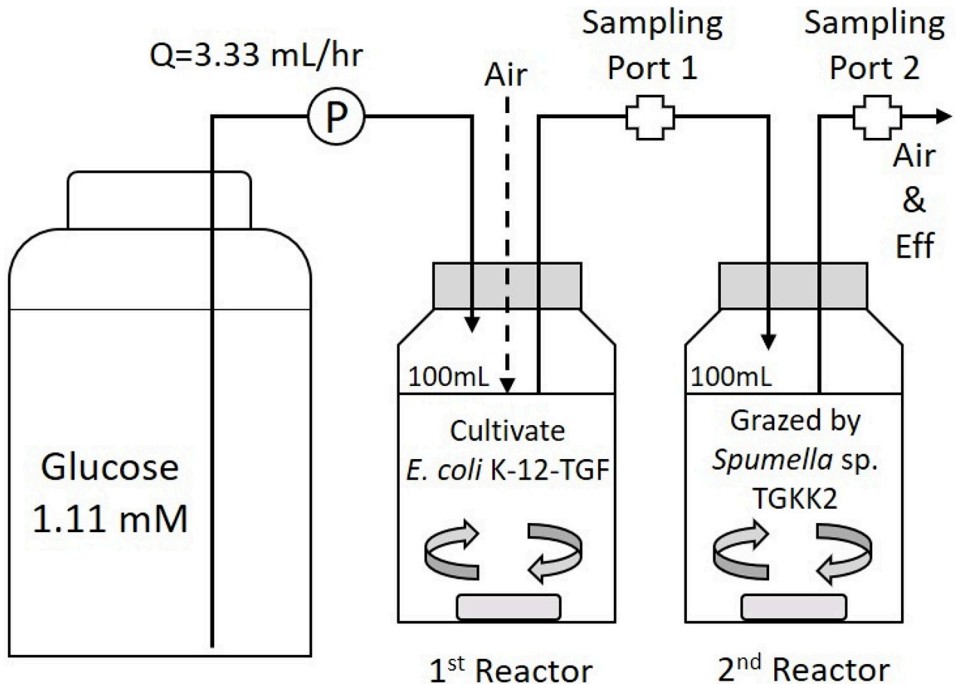

**Fig 2. Continuous culture device.** It consists of two complete mixed reactors connected in series and operated at $20 \pm 1°C$.

K-12-KRF were prepared, and *Spumella* sp. TGKK2 was added to these tubes at the concentration of $10^3$ protist-cells $mL^{-1}$. *Spumella* sp. TGKK2 concentrations were measured and adjusted by Toma's hemocytometer. Four weeks later, the protist-untouched *E. coli* K-12-TGF was added to tubes at the same level as the residual bacterium to assess the predation process. Similar experiments were also performed by exchanging the bacterial strains.

In experiments with continuously cultured bacteria, a steady state was assumed when the number of residual bacteria and protists in the effluents became constant 22 days after the operation. Then, the sample waters were collected for the predation experiments. Two grazing experiments were conducted. In the first experiment, *E. coli* K-12-TGF in the effluent of the first reactor of the device (Fig 2) was mixed with batch-cultured *E. coli* K-12-KRF, adjusting the two bacterial species at the same level. Then *Spumella* sp. TGKK2 was added to this tube at the concentration of $10^3$ protist-cells $mL^{-1}$. In the second experiment, the batch-cultured *E. coli* K-12-KRF was added to the effluent of the second reactor (Fig 2) containing residual *E. coli* K-12-TGF and *Spumella* sp. TGKK2. These experiments were triplicated.

In predation studies, 15-mL volume sterile polypropylene tubes were used and sampled periodically. Predation experiments were conducted using a shaking incubator, with shaking set at 180 rpm at 20°C. The number of prey bacteria that survived was determined by CFU on LB agar with Tc or Km. Also, the fluorescent color of colonies was often checked by the fluorescence microscope (Axio Imager.M2, ZEISS Co.) equipped with the digital color camera (AxioCam MRc5, ZEISS Co.). The detection of the ZsGreen protein was performed using Filter Set 38HE (ZEISS Co.), while the detection of the tdTomato protein was carried out using Filter Set 43HE (ZEISS Co.).

## Results

### Prey bacteria

Two types of transformants, *E. coli* K-12-TGF and K-12-KRF, were developed for predatory experiments. These strains were grown on LB agar plates containing Tc or Km overnight at 37°C. Microscopic images of these strains were taken as shown in Fig 3. The upper images (Fig 3A and 3C) are phase-contrast microscope photographs, while the lower images (Fig 3B and 3D) are fluorescent microscope photographs. Two strains showed their specific fluorescent colors.

### Water quality of continuous culture

Water samples were collected from Day 22 to Day 31 during steady state. The first reactor effluent was collected from sampling port 1, and the second reactor effluent was collected from sampling port 2 (Fig 2). Water samples were collected four times (Days 22, 25, 28, and 31) to measure water quality. Glucose concentration was lower than the detection limit (3.6 mg L$^{-1}$) on Day 22 and was not measured after that. The means and standard deviations of the water

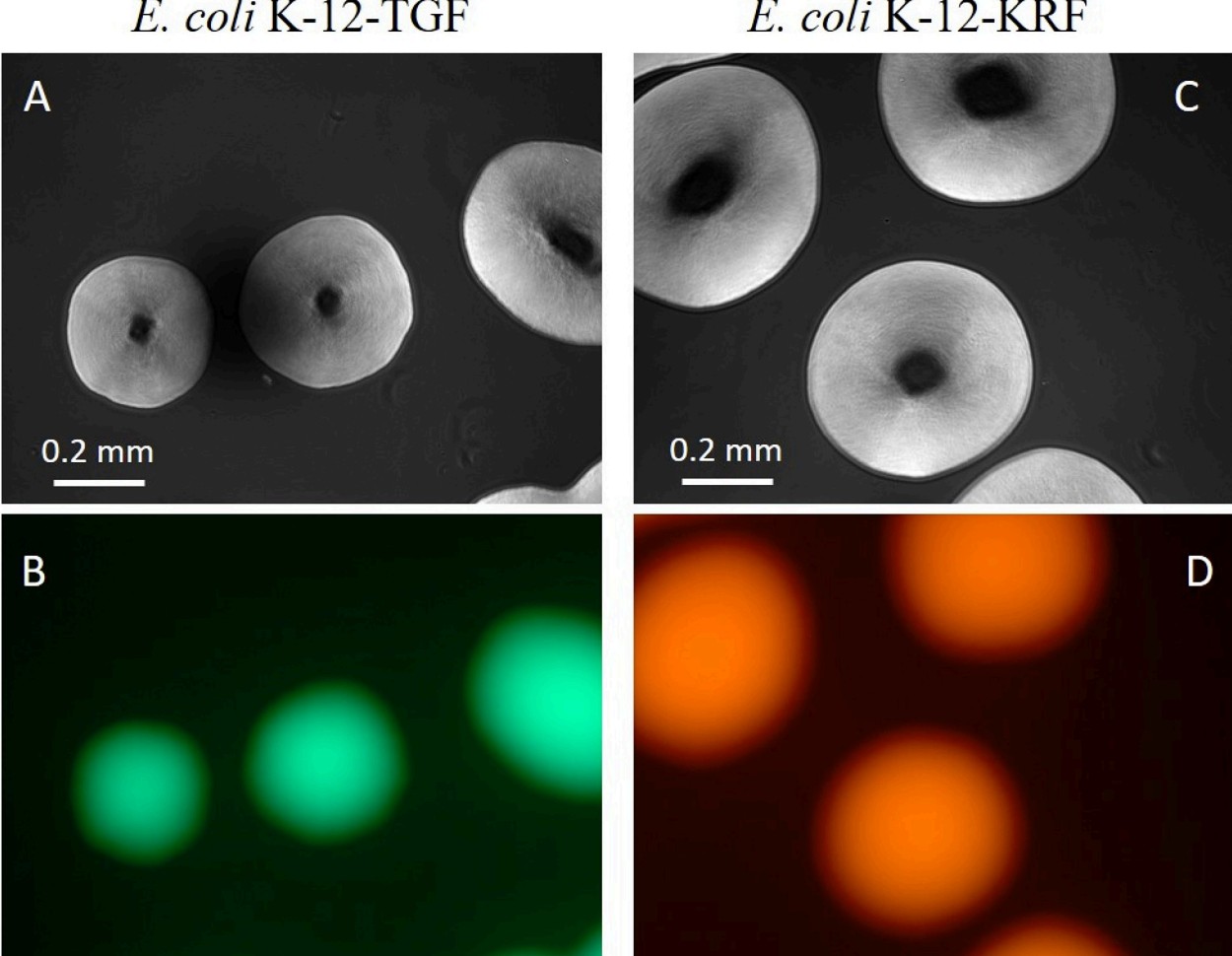

**Fig 3. Microscopic images of *E. coli* K-12-TGF and K-12-KRF.** (A) Phase-contrast picture of K-12-TGF. (B) Fluorescent picture of K-12-TGF. (C) Phase-contrast picture of K-12-KRF. (D) Fluorescent picture of K-12-KRF.

**Table 2. Water quality of continuous flow reactor effluents under presumable steady state.**

| Measurement items | 1st Reactor Eff | 2nd Reactor Eff |
|---|---|---|
| pH | 6.40±0.03 | 6.73±0.04 |
| All TOC (mg L$^{-1}$) | 20.96±0.88 | 19.77±0.42 |
| [a]Supernatant TOC (mg L$^{-1}$) | 9.94±0.34 | 17.66±0.72 |
| [b]Suspended TOC (mg L$^{-1}$) | 11.02±0.59 | 2.11±0.52 |
| Bacterial number (CFU mL$^{-1}$) | 3.54±0.19 E+08 | 1.89±0.81 E+06 |
| Protistan number (protist-cells mL$^{-1}$) | - | 8.56±0.77 E+05 |
| Conversion rate (CFU protist-cell$^{-1}$) | - | 417±58 |

[a]Supernatant TOC: supernatant of sample after centrifuge (11,100 x g, 10 min, 4°C)
[b]Suspended TOC: calculated value (All TOC minus Supernatant TOC)

quality measurements are shown in Table 2. The original data is shown in S1 Table. The pH was 6.40 for the first vessel effluent and 6.73 for the second vessel effluent. A slight increase was observed. Comparison of the effluent in the first and second vessels for TOCs showed some differences. All TOC slightly decreased from 20.95 to 19.77 mg L$^{-1}$. Supernatant TOC increased from 9.94 to 17.66 mg L$^{-1}$. Suspended TOC decreased from 11.01 to 2.11 mg L$^{-1}$. The bacterial number in the first reactor reached $3.54 \times 10^8$ CFU mL$^{-1}$ upon growth with glucose. Whereas in the second reactor, protistan predation resulted in $1.89 \times 10^6$ CFU mL$^{-1}$, which decreased to approximately 0.5% of the first reactor value. This bacterial decrease resulted in protists increasing to approximately $8.56 \times 10^5$ protist-cells mL$^{-1}$, with a 417 CFU protist-cell$^{-1}$ conversion rate.

## Grazing resistance in batch-cultured bacteria

After 28 days of the start of the predation, residual bacterial counts reached approximately $10^6$ CFU mL$^{-1}$ regardless of the initial bacterial level, as shown in Fig 4B–4D. Then, the residual *E. coli* K-12- KRFs in Tubes 2, 3, and 4 were supplemented with the protist-untouched *E. coli* K-12- TGF of Tube 1 (Fig 4A), adjusting at a concentration of $4.0 \times 10^6$ CFU mL$^{-1}$ (arrow on Day 28 in Fig 4B–4D). Bacterial counts were determined one week later and showed little changes in bacterial counts of *E. coli* K-12-KRFs remaining via predation but rapid decreases in newly added *E. coli* K-12-TGFs to approximately 0.8% of the initial value (shaded period of Fig 4B–4D). Then, the combination of the strains was reversed to prove that this phenomenon is not due to the native nature of *E. coli* K-12-TGF and K-12-KRF, as shown in Fig 4F–4H. The reduction in *E. coli* K-12-TGF 28 days after initiating the predation experiment was similar to the experiment shown in Fig 4B–4D and reached approximately $10^6$ CFU mL$^{-1}$ regardless of the initial concentration. The protist-untouched *E. coli* K-12-KRF of Tube 5 (Fig 4E) was then added at a concentration of $4.0 \times 10^6$ CFU mL$^{-1}$ in three tubes, Tubes 6, 7 and 8 (Fig 4F–4H). *E. coli* K-12-KRF decreased significantly after one week to about 0.5% at the time of addition (shaded period of Fig 4F–4H). On the other hand, each of *E. coli* K-12-TGFs with a predation history was less grazed, showing no significant change in bacterial count. The fluorescent colors of bacterial colonies were often checked, and the expected colors were consistently observed. The numerical data of Fig 4 is shown in S2 Table.

## Grazing-resistance in continuous-cultured bacterium

The device shown in Fig 2 was operated to obtain continuous-cultured and continuous-grazed *E. coli* K-12-TGF. First, *E. coli* K-12-TGF from the first reactor was mixed with batch-cultured *E. coli* K-12-KRF and *Spumella* sp. TGKK2. Fig 5A shows the result of the grazing experiment

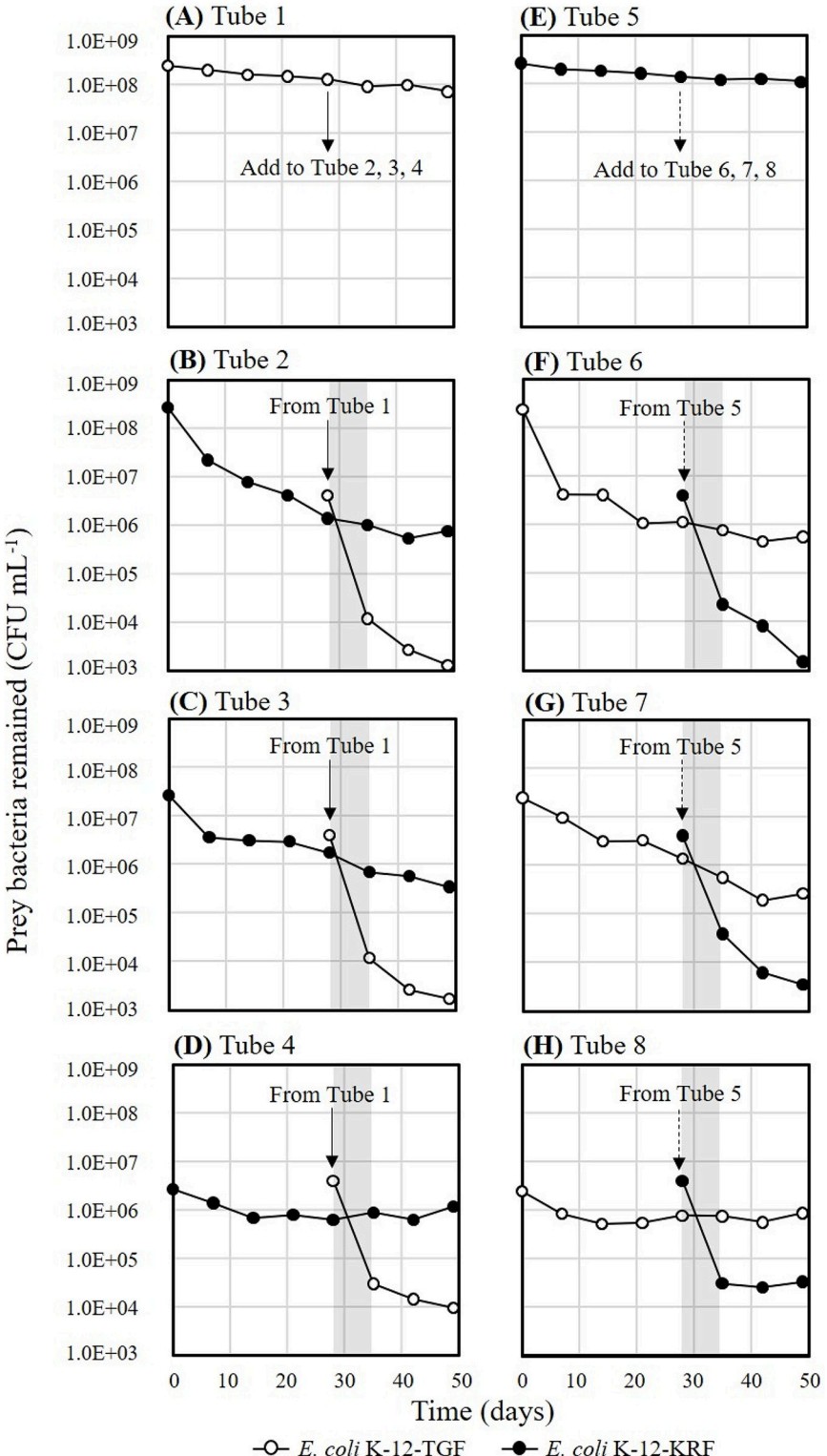

**Fig 4. Predation assessment of batch-cultured bacteria.** Open symbols are *E. coli*-K12-TGF, and closed symbols are *E. coli*-K12-KRF. (A) *E. coli*-K12-TGF was cultivated without predation. (B, C, D) *E. coli*-K12-KRFs were grazed by *Spumella* sp. TGKK2 for 28 days, then K12-TGF in Tube 1 was added to each tube. The combination was exchanged. (E) *E. coli*-K12-KRF was cultivated without predation. (F, G, H) *E. coli*-K12-TGFs were grazed by *Spumella* sp. TGKK2 for 28 days, then K12-KRF in Tube 5 was added to each tube.

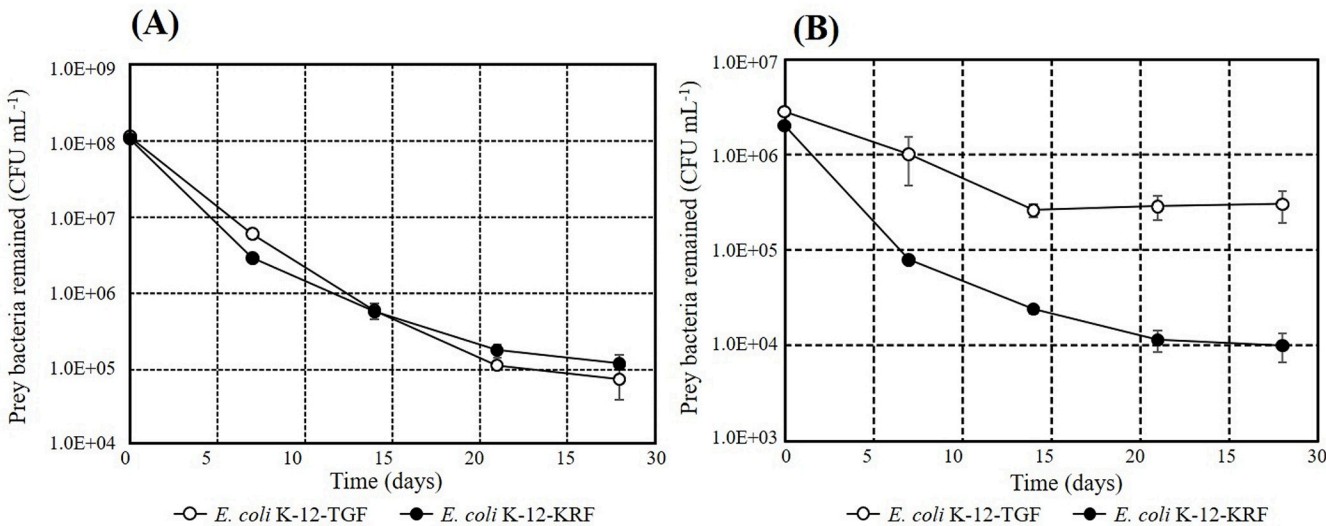

**Fig 5. Predation assessment of continuous-cultured bacteria.** (A) Predation assessment of continuous-cultured bacterium and batch-cultured bacteria. Open symbols are *E. coli*-K12-TGF, and closed symbols are *E. coli*-K12-KRF. *E. coli*-K12-TGF from the first reactor of the continuous device and batch-cultured *E. coli*-K12-KRF were grazed. (B) Predation assessment of continuously predated bacteria and batch-cultured bacteria. Open symbols are *E. coli*-K12-TGF, and closed symbols are *E. coli*-K12-KRF. *E. coli*-K12-TGF from the second reactor of the continuous device and batch-cultured *E. coli*-K12-KRF were grazed. Plots are the average of triplicate experiments. Error bars represent the standard deviation.

for 28 days. Approximately $10^8$ CFU mL$^{-1}$ of the initial bacterial count was reduced to 6% or less after 7 days and 0.6% or less after 14 days in all cases. Also, the final count was settled at approximately $1 \times 10^5$ CFU mL$^{-1}$. The process of predation-induced reduction and the final attained levels did not differ markedly between *E. coli* K-12-TGF from continuous culture and *E. coli* K-12-KRF from batch culture. Second, a comparative predation experiment was performed by adding *E. coli* K-12-KRF to the effluent of the second tank containing *E. coli* K-12-TGF to the same concentration level as shown in Fig 5B. *Spumella* sp. TGKK2 was contained in the effluent at the concentration of $8.72 \times 10^5$ protist-cells mL$^{-1}$. Fig 5B shows the time course of residual bacterial counts. The rate of reduction in bacterial counts was slower in *E. coli* K-12-TGF with a history of protistan predation than *E. coli* K-12-KRF with no protistan contact. There were also significant differences in the final residual bacterial counts. *E. coli* K-12-TGF survived preferentially, showing more than 20-fold greater counts compared to *E. coli* K-12-KRF. The fluorescent colors of bacterial colonies were often checked and always showed the expected colors. The numerical data of Fig 5 is shown in S3 Table.

## Comparison of the reduction rate of prey bacteria

It was clarified from the above experimental result that predation resistance was acquired in the history of coexistence with the protist. For further investigation, the rate of reduction in bacterial counts was calculated from the results of Figs 4 and 5. The rates were compared by the reaction rate constant k, assuming a first-order reaction, as shown in Fig 6. The calculated k values were shown by 8 bars. The 1st and 2nd bars are the spontaneous rates of decline for *E. coli* K-12-TGF and *E. coli* K-12-KRF, calculated by least squares from the entire measurements from Fig 4A (Tube 1) and 4D (Tube 5), respectively, with the k values very small as −0.024 day$^{-1}$ and −0.017 day$^{-1}$. The coefficient of determination ($R^2$) for each was 0.98 and 0.94, respectively. The 3rd and 4th bars were calculated from the results of Fig 4B–4D. The calculated k value of the 3rd bar from day 28 to day 35, on *E. coli* K-12-KRF with predation histories in three tubes (Tubes 2, 3, and 4), yielded an average of -0.042 day$^{-1}$ with a standard deviation

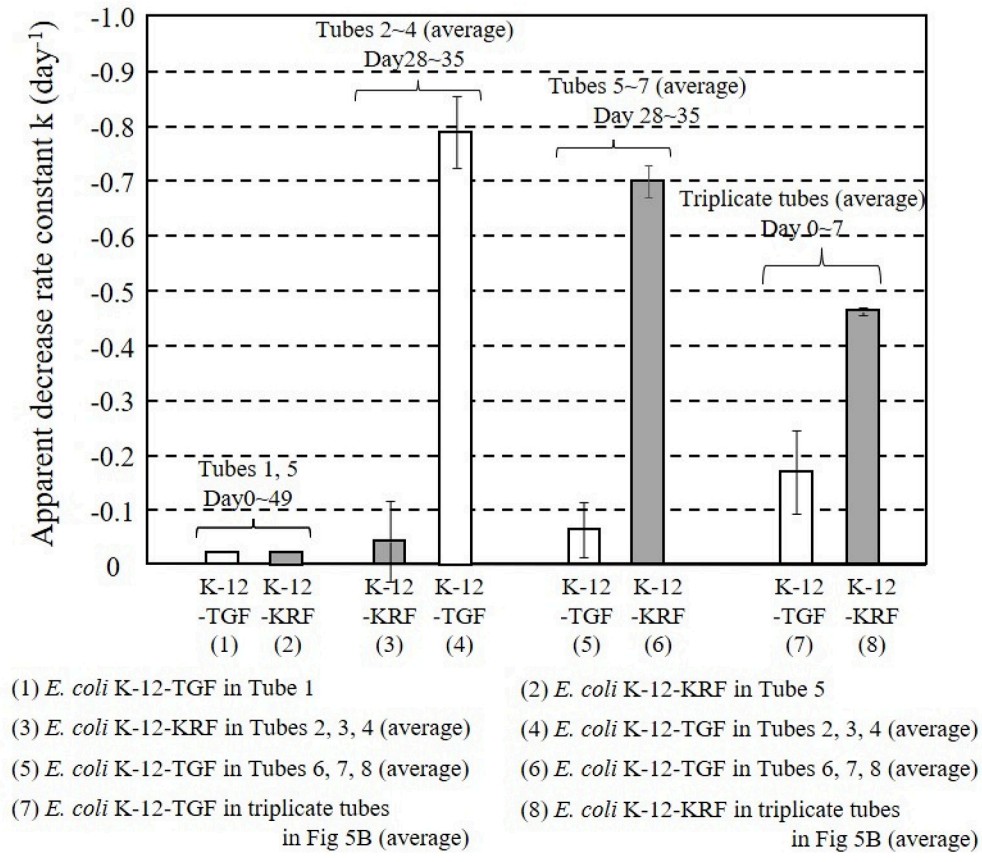

**Fig 6. Comparison of reduction rates of prey bacteria in predation assessments.** (1) and (2) show decreasing rates without grazing. Second pair (3) (4), third pair (5) (6), and fourth pair (7) (8) show the effect of predation histories. Error bars represent the standard deviation of the decrease rates (k values).

of 0.074 (hereinafter expressed as mean ± standard deviation). The k value of the 4th bar was −0.790±0.065 day$^{-1}$ for *E. coli* K-12-TGF without any predation history in the same three tubes. This k value was about 18-fold greater compared to those with predation history. A two-tailed t-test with a 95% confidence interval was conducted to evaluate the significant difference between these two k values. The P-value of the t-test was 0.0011 (<0.05), and the significant difference was confirmed. Similar phenomena were also shown when combinations were reversed, as shown in the 5th and 6th bars. The k value of the 5th bar was −0.063±0.051 day$^{-1}$ for *E. coli* K-12-TGF with predation histories in the three tubes (Tubes 6, 7, and 8) in Fig 4. The k value of the 6th bar was −0.699 ±0.029 day$^{-1}$ for *E. coli* K-12-KRF without predation history. This k value was about 10-fold greater compared to those with predation history. The calculated P-value of the t-test was 0.0060 (<0.05), and the significant difference was confirmed. In the predation experiment with the continuously cultured bacterium illustrated in Fig 5B, the initial 7 day k value for *E. coli* K-12-KRF without a history of predation was -0.462 ± 0.007 day$^{-1}$, indicated by the 8th bar. In contrast, for *E. coli* K-12-TGF with a predation history in the same tube, the k value became -0.168± 0.076 day$^{-1}$, as shown by the 7th bar, about one-third of the above k value. The calculated P-value of the t-test was 0.038 (<0.05), confirming the significant difference. The numerical data of Fig 6 is shown in S4 Table.

## Discussion

The experimental findings presented in Figs 4–6 suggested that the bacteria remaining after predation are less likely to be predated by the protist compared to the protist-untouched bacteria. In Fig 4, some predation resistance was likely to be acquired during the 28-day coexistence with protists. The same type of grazing resistance was also developed in 30 h of HRT with the continuous reactor, as shown in Fig 5. Furthermore, the comparison of k values (Fig 6) with the t-test confirmed the development of some grazing resistance in *E. coli* K-12 coexisting with the protist.

Table 2 summarizes the results of the water quality analysis of the effluents from the serial device's first and second vessels. *E. coli* K-12-TGF grown in the first reactor was predated by *Spumella* sp. TGKK2 in the second reactor. The initial bacterial number of $3.54 \times 10^8$ CFU mL$^{-1}$ was reduced to less than 1% (= $1.89 \times 10^6$ CFU mL$^{-1}$). When similar tests were conducted using *Cupriavidus* sp. KN1-TGF previously, the bacterium at $10^8$ CFU mL$^{-1}$ level was also reduced to $10^6$ CFU mL$^{-1}$ level by predation [8]. *E. coli* K-12-TGF was considered to have similar predatory properties. On the other hand, the conversion rate of 417 CFU protist$^{-1}$ was approximately twice as high as 224–231 CFU protist$^{-1}$ for *Cupriavidus* sp. KN1-TGF [8].

In this study, TOC was categorized into three: all TOC, supernatant TOC, and suspended TOC. The supernatant TOC of 9.94 mg L$^{-1}$ in the first reactor effluent was not degraded by *E. coli* K-12-TGF. Therefore, this supernatant TOC should be mainly unchanged in the second reactor. On the other hand, the suspended TOC of 11.01 mg L$^{-1}$ consisting of bacterial cells was decreased by predation because more than 99% of the number of bacteria was decreased. The decreased value of 8.90 mg L$^{-1}$ (from 11.01 to 2.11 mg L$^{-1}$) in suspended TOC was considered to be converted to $CO_2$ (1.18 mg L$^{-1}$: all TOC decreased), and the increased supernatant TOC (7.72 mg L$^{-1}$). Whether or not this increased supernatant TOC produced by grazing affects the formation of grazing resistance is currently unknown.

A comparison of the predation-induced reduction rates of prey bacteria was conducted with the reaction rate constant k (day$^{-1}$) calculated by assuming a first-order reaction (Fig 6). The differences in the rate of decline between those with and without predation history were significant in both batch and continuous cultured *E. coli* strains. In this figure, the lower the k value, the more grazing resistant. *E. coli* K-12-TGF contacted with the protist for 28 days in tubes showed more grazing resistance than *E. coli* K-12-TGF exposed to the protist for 30 hours in the vessel. These two k values, $-0.063 \pm 0.051$ and $-0.168 \pm 0.076$ day$^{-1}$, went to the t-test. However, the calculated P-value was 0.324 (>0.05) which did not show a significant difference. The importance of the protist coexisting time should be addressed in more detail in a future study.

Interesting results were also obtained on the relationship between protozoa and bacteria. Fig 4B–4D, 4F–4H show that the first added protest-untouched bacteria decreased to approximately $10^6$ CFU mL$^{-1}$ by Day 28, irrespective of the initial bacterial concentrations. However, the second added protest-untouched bacteria on Day 28 have dropped to much lower concentrations. Thus, once a population of predation-resistant bacteria was formed, the subsequently added grazing-sensitive bacteria was selectively predated to very low concentrations. Similar experimental results were also observed in our previous study [7]. In that study, river water containing $10^6$ cells mL$^{-1}$ of indigenous bacteria and some indigenous protists was used. *Cupriavidus necator* KT1 of $5 \times 10^5$ cells mL$^{-1}$ (final concentration) was added to the river water, and the bacterial consortium was analyzed by Terminal Restriction Fragment Length Polymorphisms (T-RFLP). The T-RF peak of *C. necator* KT1 was not detected 5 days later. It meant that *C. necator* KT1 decreased to less than $10^4$ cells mL$^{-1}$ levels, the T-RF peak detection limit in that experiment. On the other hand, almost no changes were observed in the

community structure of indigenous bacteria. Thus, it can be interpreted that indigenous bacteria in rivers have some predation resistance, and the addition of protist-untouched bacteria to them causes the same events as after Day 28 in Fig 4. This phenomenon will become a significant problem in releasing some cultured bacteria as bioaugmentation into contaminated groundwater where indigenous grazing-resistant bacteria exist. At this point, the underlying mechanism of this phenomenon remains unknown. More studies are needed to understand the mechanism.

Many reviews exist regarding grazing resistance [14–16], and possible mechanisms have also been proposed from the microbial ecology viewpoint. In the review [16], microbial anti-predator strategies were classified into the following seven types: 1) Size Reduction, 2) Cell wall structure, surface potential, 3) Morphology, filamentation, 4) Exopolymer formation, 5) Communication, 6) Toxin release, 7) Motility pattern. In our experiments, grazing resistance has been observed in common bacteria, such as *Cupriavidus* sp. and *E. coli*, with no noticeable shape and motility changes in microscopic observations. And, it is unlikely to be the effect of the chemical substance that suppresses the predation because neither bacterium is known to produce such substances. Changes in cell surface characteristics are possibly considered. The former research evaluated the effects of cell surface hydrophobicity and charge on predation [17]. However, the observed grazing resistance was not as significant as the level of predation resistance we observed in this research. Thus, similar results were not found in previously published papers on bacterial grazing resistance caused by protozoan contact. More investigation is needed to explain the grazing resistance observed in this study.

Our studies indicate that transformants of *E. coli* K-12 from human feces developed grazing resistance through contact with the protist, similar to *Cupriavidus* sp. KN1 from the natural environment [8]. Thus, predation resistance acquisition may be a common phenomenon with a wide range of bacteria rather than a characteristic restricted to a particular bacterium. In the study, no mutant strain that poses permanent grazing resistance was identified, which is similar to the study on *Cupriavidus* sp KN1 [8]. Therefore, a possible explanation for predation resistance is due to changes in the properties of bacteria caused by protistan contacts. If it is true, some genes that act through protistan contacts may be involved. Hence, future research will focus on exploring genes related to acquiring grazing resistance.

## Supporting information

**S1 Fig. Growth of *E. coli* strains on LB agar plates with and without antibiotics.** LB agar, LB agar containing 5 mg L$^{-1}$ of Tc, and LB agar containing 50 mg L$^{-1}$ of Km are used. The left colonies are *E. coli* K-12. The center colonies are *E. coli* K-12-TGF. The right colonies are *E. coli* K-12-KRF. All strains were transferred to medium and incubated overnight at 37°C.
(TIF)

**S1 Table. Original data in Table 2.** The means and the standard deviations were calculated from these data.
(PDF)

**S2 Table. Numerical data of Fig 4.** The graph was made from these data.
(PDF)

**S3 Table. Numerical data of Fig 5.** The graph was made from these data.
(PDF)

**S4 Table. Numerical data of Fig 6.** The graph was made from these data.
(PDF)

## Author Contributions

**Conceptualization:** Kanji Nakamura.

**Data curation:** Kanji Nakamura, Keisuke Miyauchi.

**Formal analysis:** Kanji Nakamura.

**Funding acquisition:** Kanji Nakamura.

**Investigation:** Kanji Nakamura, Keisuke Miyauchi.

**Methodology:** Kanji Nakamura, Keisuke Miyauchi.

**Project administration:** Kanji Nakamura.

**Supervision:** Kanji Nakamura.

**Writing – original draft:** Kanji Nakamura.

**Writing – review & editing:** Keisuke Miyauchi.

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
