## [Decision Letter · Decision Letter 0]

2 Nov 2023

PONE-D-23-32208

Grazing resistance developed in Escherichia coli K-12 during coexistence with a bacterivorous protist

PLOS ONE

Dear Dr. Nakamura,

Thank you for submitting your manuscript to PLOS ONE. After careful consideration, we feel that it has merit but does not fully meet PLOS ONE’s publication criteria as it currently stands. Therefore, we invite you to submit a revised version of the manuscript that addresses the points raised during the review process.

We look forward to receiving your revised manuscript.

Kind regards,

Amitava Mukherjee, ME, Ph.D.

Academic Editor

PLOS ONE

Journal Requirements:

   "JSPS KAKENHI Grant Numbers 20K21023"

Reviewers' comments:

Reviewer's Responses to Questions

**Comments to the Author**

1. Is the manuscript technically sound, and do the data support the conclusions?

Reviewer #1: No

Reviewer #2: Yes

Reviewer #3: Yes

2. Has the statistical analysis been performed appropriately and rigorously? 

Reviewer #1: No

Reviewer #2: N/A

Reviewer #3: Yes

3. Have the authors made all data underlying the findings in their manuscript fully available?

Reviewer #1: Yes

Reviewer #2: Yes

Reviewer #3: Yes

4. Is the manuscript presented in an intelligible fashion and written in standard English?

Reviewer #1: Yes

Reviewer #2: Yes

Reviewer #3: Yes

5. Review Comments to the Author

Reviewer #1: Dear Author

Thank you for your manuscript submission. Although this type of study is not new (there are a wide range of examples e.g., the bacteriovorous bacterium of Bdellovibrio etc.), it can be interesting in this field of research. There are two important points that should be included in the present manuscript:

1. It is necessary to add statistical analyses to show the efficacy of the obtained results through the statistical language.

2. You have mentioned "two transformants were generated from E. coli K12 for grazing experiments. One was E. coli K-12-TGF, which possesses tetracycline resistance and green fluorescence. The other was E. coli K-12-KRF with kanamycin resistance and red fluorescence".

It is necessary to add the original figures relating to fluorescent colors (Red and Green) in this regard.

3. It is important to interpret the statistical correlations in Discussion section.

4. No effective conclusion has been presented.

Reviewer #2: Your manuscript titled "Grazing resistance developed in Escherichia coli K-12 during coexistence with a

bacterivorous protist" is a significant contribution to bioremediation processes. Your hypothesis is that indigenous bacteria can acquire some grazing-resistant traits by coexisting with protists in the natural environment. To verify the same you performed some experiments by using a bacterium, Cupriavidus sp. KN1 and a purified bacterivorous protist, Spumella sp. TGKK2

A development of grazing resistance in Escherichia coli K-12 was examined in the presence of a bacterivorous protist, Spumella sp. TGKK2. Two transformants were generated from E. coli K12 for grazing experiments. One was E. coli K-12-TGF, which possesses tetracycline resistance and green fluorescence and the other was E. coli K-12-KRF with kanamycin resistance and red fluorescence. First, you added protist-untouched E. coli K-12-KRF to protist-touched residual E. coli K-12-TGF that had been attacked by Spumella sp. TGKK2 in a batch test. Then the survivability of the respective strains was investigated. Consequently, E. coli K-12-KRF was predated preferentially. On the other hand, E. coli K-12-TGF in the same tube was less predated, indicating some grazing resistance. Also, a continuous culture device supplied with a glucose-containing medium as a substrate was operated. The device connected two complete mixed reactors in series. E. coli K-12-TGF was cultivated in the first reactor, and then grown E. coli K-12-TGF was predated by Spumella sp. TGKK2 in the second reactor. The effluent in the second reactor containing residual E. coli K-12-TGF and Spumella sp. TGKK2 was supplemented with batch-cultured E. coli K-12-KRF. Consequently, it was confirmed that bach-cultured E. coli K-12-KRF never exposed to protist was predated preferentially. These findings reveal that E. coli K12 acquires some predation resistance through coexistence with the bacterivorous protist.

The manuscript is correctly developed with a detailed description of the Materials and Methods applied, which allowed them to achieve robust results.

The only weakness that I would like to comment on is the lack of a conclusion after carrying out the present study since they only mention "However, the observed grazing resistance was not as significant as the level of predation resistance we observed in this research. More investigation is needed to explain the grazing resistance observed in this study."

Reviewer #3: The research work sets clear objectives, the experiments seem solid, although there is doubt as to why the inoculum concentration is not standardized, it is not explained why the concentrations of 5 to 50 between E coli TGF and KRF.

The statistical results are expressed in the graphs, but are not described within the text.

If the reasons why it is not standardized and references are included, as well as the statistical data described in the text, the work will be even more robust.

6. PLOS authors have the option to publish the peer review history of their article (what does this mean?). If published, this will include your full peer review and any attached files.

Reviewer #1: **Yes: **Payam BEHZADI

Reviewer #2: No

Reviewer #3: **Yes: **Jorge Angel Almeida Villegas

---

## [Author Response · Author response to Decision Letter 0]

28 Jan 2024

Review Comments to the Author

Reviewer #1: Dear Author

Thank you for your manuscript submission. Although this type of study is not new (there are a wide range of examples e.g., the bacteriovorous bacterium of Bdellovibrio etc.), it can be interesting in this field of research. There are two important points that should be included in the present manuscript:

1. It is necessary to add statistical analyses to show the efficacy of the obtained results through the statistical language.

(Answer 1)

Old Fig 4 was changed as shown new Fig 5. Also Statistical analyses were added (L273-295). Mainly, t-tests were conducted. The corrected parts are marked in red.

2. You have mentioned "two transformants were generated from E. coli K12 for grazing experiments. One was E. coli K-12-TGF, which possesses tetracycline resistance and green fluorescence. The other was E. coli K-12-KRF with kanamycin resistance and red fluorescence".

It is necessary to add the original figures relating to fluorescent colors (Red and Green) in this regard.

(Answer 2)

In RESULTS, “Prey bacteria” was added (L175-185). The corrected parts are marked in red. Also microscopic pictures were added as new Fig 3. 

3. It is important to interpret the statistical correlations in Discussion section.

(Answer 3)

A t-test was conducted and the result was discussed (L307-309). The corrected parts are marked in red.

4. No effective conclusion has been presented.

(Answer 4)

The last part of the discussion was modified to show a conclusion (L372-384). The corrected parts are marked in red.

Reviewer #2: Your manuscript titled "Grazing resistance developed in Escherichia coli K-12 during coexistence with a bacterivorous protist" is a significant contribution to bioremediation processes. Your hypothesis is that indigenous bacteria can acquire some grazing-resistant traits by coexisting with protists in the natural environment. To verify the same you performed some experiments by using a bacterium, Cupriavidus sp. KN1 and a purified bacterivorous protist, Spumella sp. TGKK2

A development of grazing resistance in Escherichia coli K-12 was examined in the presence of a bacterivorous protist, Spumella sp. TGKK2. Two transformants were generated from E. coli K12 for grazing experiments. One was E. coli K-12-TGF, which possesses tetracycline resistance and green fluorescence and the other was E. coli K-12-KRF with kanamycin resistance and red fluorescence. First, you added protist-untouched E. coli K-12-KRF to protist-touched residual E. coli K-12-TGF that had been attacked by Spumella sp. TGKK2 in a batch test. Then the survivability of the respective strains was investigated. Consequently, E. coli K-12-KRF was predated preferentially. On the other hand, E. coli K-12-TGF in the same tube was less predated, indicating some grazing resistance. Also, a continuous culture device supplied with a glucose-containing medium as a substrate was operated. The device connected two complete mixed reactors in series. E. coli K-12-TGF was cultivated in the first reactor, and then grown E. coli K-12-TGF was predated by Spumella sp. TGKK2 in the second reactor. The effluent in the second reactor containing residual E. coli K-12-TGF and Spumella sp. TGKK2 was supplemented with batch-cultured E. coli K-12-KRF. Consequently, it was confirmed that batch-cultured E. coli K-12-KRF never exposed to protist was predated preferentially. These findings reveal that E. coli K12 acquires some predation resistance through coexistence with the bacterivorous protist.

The manuscript is correctly developed with a detailed description of the Materials and Methods applied, which allowed them to achieve robust results.

The only weakness that I would like to comment on is the lack of a conclusion after carrying out the present study since they only mention "However, the observed grazing resistance was not as significant as the level of predation resistance we observed in this research. More investigation is needed to explain the grazing resistance observed in this study."

(Answer 5)

The last part of the discussion was modified to show a conclusion (L372-384). The corrected parts are marked in red.

Reviewer #3: The research work sets clear objectives, the experiments seem solid, although there is doubt as to why the inoculum concentration is not standardized, it is not explained why the concentrations of 5 to 50 between E coli TGF and KRF.

(Answer 6)

Each of Tc and Km had its optimum concentration to distinguish E. coli TGF from KRF. The growth of these strains on LB containing 5 mg L-1 of Tc or 50 mg L-1 of Km was shown in S1 Fig., which was newly added.

The statistical results are expressed in the graphs, but are not described within the text.

If the reasons why it is not standardized and references are included, as well as the statistical data described in the text, the work will be even more robust.

(Answer 7)

Text data were added (L274-295). The corrected parts are marked in red.

---

## [Decision Letter · Decision Letter 1]

19 Feb 2024

Grazing resistance developed in Escherichia coli K-12 during coexistence with a bacterivorous protist

PONE-D-23-32208R1

Dear Dr. Nakamura,

We’re pleased to inform you that your manuscript has been judged scientifically suitable for publication and will be formally accepted for publication once it meets all outstanding technical requirements.

Kind regards,

Amitava Mukherjee, ME, Ph.D.

Academic Editor

PLOS ONE

Additional Editor Comments (optional):

Reviewers' comments:

Reviewer's Responses to Questions

**Comments to the Author**

1. If the authors have adequately addressed your comments raised in a previous round of review and you feel that this manuscript is now acceptable for publication, you may indicate that here to bypass the “Comments to the Author” section, enter your conflict of interest statement in the “Confidential to Editor” section, and submit your "Accept" recommendation.

Reviewer #1: All comments have been addressed

Reviewer #2: All comments have been addressed

2. Is the manuscript technically sound, and do the data support the conclusions?

Reviewer #1: Yes

Reviewer #2: Yes

3. Has the statistical analysis been performed appropriately and rigorously? 

Reviewer #1: Yes

Reviewer #2: Yes

4. Have the authors made all data underlying the findings in their manuscript fully available?

Reviewer #1: Yes

Reviewer #2: Yes

5. Is the manuscript presented in an intelligible fashion and written in standard English?

Reviewer #1: Yes

Reviewer #2: Yes

6. Review Comments to the Author

Reviewer #1: Dear Author

Thank you for the related revisions. I believe that your manuscript can be published in present form.

Reviewer #2: With the modifications and clarifications made by the authors, the manuscript entitled "Grazing resistance developed in Escherichia coli K-12 during coexistence with a bacterivorous protist" is ready to be accepted for publication in Plos One.

7. PLOS authors have the option to publish the peer review history of their article (what does this mean?). If published, this will include your full peer review and any attached files.

Reviewer #1: **Yes: **Payam BEHZADI

Reviewer #2: **Yes: **Nora Mestorino

---

## [Editor Report · Acceptance letter]

1 Apr 2024

PONE-D-23-32208R1 

PLOS ONE

Dear Dr. Nakamura, 

I'm pleased to inform you that your manuscript has been deemed suitable for publication in PLOS ONE. Congratulations! Your manuscript is now being handed over to our production team.

Kind regards, 

on behalf of

Professor Dr. Amitava Mukherjee 

Academic Editor

PLOS ONE